# GenST: A Generative Cross-Modal Model for Predicting Spatial Transcriptomics from Histology Images

**Ruby Wood**
**Yang Hu**
**Jens Rittscher**
**Bin Li**                                          bin.li@eng.ox.ac.uk
*Big Data Institute, University of Oxford, UK*

**Editor:** COMputational PAthologY and multimodaL data workshop (COMPAYL)

## Abstract

Spatial transcriptomics is used to identify gene expression levels in certain locations across a tissue sample, preserving important spatial information in cancerous tissue samples for downstream clinical decision making. However, this technology is currently too expensive to be used in a routine clinical pathways. On the other hand, digital images of haematoxylin and eosin stained histology slides are routinely generated from tissue biopsy samples. Here, we develop a generative cross-modal method to predict spatial transcriptomics from histology images by aligning the latent space of two VQ-VAEs for each modality. We benchmark our approach on multiple sequencing technologies (Visium and ST) and cancer types (breast, brain, spinal cord and skin) from two public datasets, using 142 slides with 820,407 spots from STImage-1K4M (Chen et al., 2024a) and 568 slides with 254,812 spots from HEST-1k (Jaume et al., 2024). Across the resulting cohorts, our model achieves superior performance to state-of-the-art models in half, whilst providing an interpretable framework for understanding which genetic expressions of a cancer tumour can be captured from the morphology observed in corresponding locations of the histology image.

**Keywords:** computational pathology, spatial transcriptomics, multimodal AI

## 1 Introduction

With the recent progression of spatial transcriptomics technology, this data modality has become more widely available in both the public and private domains. Spatial transcriptomics allows the user to read expression levels of multiple specified genes from biopsy tissues, whilst also recording the corresponding location in the image of such expressions. Such information can be useful to quantify the tumour microenvironment landscape and address the issue of predicting patient outcomes in the context of personalised medicine for cancer treatment (Williams et al., 2022). However, the methods for reading spatial transcriptomics are still prohibitively expensive for regular use in the clinic (Smith et al., 2024), and so our research focuses on the task of predicting these spatial gene expressions from the corresponding histology whole slide image, a routinely taken biopsy which is stained with haematoxylin and eosin (H&E) before being scanned by a microscope at high resolution.

While other approaches have been developed to predict ST expressions from the corresponding histology image (Nonchev et al., 2025; Xie et al., 2023; He et al., 2020; Yang et al., 2024; Pang et al., 2021; Zeng et al., 2022; Min et al., 2024; Shulman et al., 2025; Dawood

et al., 2021; Chen et al., 2025), ours is the first to develop such a generative AI approach to this problem, aligning the two modalities in their latent spaces while ensuring the generated outputs remain realistic. Our method simultaneously trains two Vector Quantized Variational Autoencoders (VQ-VAE) models (Van Den Oord et al., 2017) end to end for each modality, and uses additional loss terms to align the latent space of the two generative models and ensure that the genes can be generated from the latent space of the image. Contrasting to the vanilla VAE used in Starfysh (He et al., 2025), we choose to work with discrete generative models, VQ-VAEs, for more interpretable and robust representations (Van Den Oord et al., 2017). At inference, the image encoder part of the model provides a low-dimensional cross-modal aligned representation of the image, which the gene model can decode for a corresponding prediction of the spatial gene expressions.

Our approach learns a low-dimensional representation at the intersection of two objectively dissimilar modalities, which can capture the underlying spatial biology and be used for downstream analysis such as clustering. A benefit of our method is that it can provide image reconstructions from the image model, to give insight into the relevant morphology captured in the shared latent space and used in the prediction of the gene expressions.

## 2 Related Work

Wang et al. (2025b) perform a thorough benchmarking of multiple methods which predict the spatial gene expression from histology images. Their analysis tests the models on breast cancer, kidney cancer and cutaneous squamous cell carcinoma, using both Spatial Transcriptomics (ST) and 10X Visium data. The results from this work demonstrate the difficulty of the task at hand, since the Pearson correlation coefficient between the true and predicted gene expressions in each spatial location rarely goes above 0.6 across the various experiments performed.

A best-performing model in this review is ST-Net (He et al., 2020). This simple approach uses a DenseNet121 neural network to encode the image, adding a classifier layer to output a prediction for each gene in the dataset. To better predict the sparse labels, they use the mean gene expression from the training set as an initial value for the bias in their final classifier layer in their prediction model, prior to training, further biasing their model towards the training domain. Another model, BLEEP, is similar to our approach in that they aim to align the latent space of the matching image and gene expression pairs (Xie et al., 2023), using a ResNet50 as their image encoder. One disadvantage of this approach, however, becomes more pronounced at inference, where they directly impute the spatial transcriptomics expression for an unseen image as an explicit computation on latents from the reference training data, resulting in a costly operation. Models mclSTExp and OmiCLIP (Min et al., 2024; Chen et al., 2025) align the latent space of the two modalities with the CLIP training approach (Radford et al., 2021), with OmiCLIP further training a downstream model for prediction of ST from histology, with modest results. DeepSpot (Nonchev et al., 2025) incorporates surrounding tissue in the histology image to help train the model. A histology foundation model (such as UNI, Chen et al., 2024b) is used as the image encoder, but for a single spot they additionally consider image features from cropped tiles contained within that spot, as well as from other neighbouring spots.

Other approaches choose to remove the raw or transformed expression counts entirely by ranking the genes in terms of the highest to lowest expressed for a single spot location, then proceed to use a natural language processing (NLP) model to encode these gene 'sentences', with each ranked gene name separated with a space (Levine et al., 2024; Min et al., 2024). While the argument is that this method deals well with batch effects, it removes the value of the gene expression which is what we aim to predict in this work.

## 3 Methods

### 3.1 Target Gene Panel Selection

To construct a unified and biologically informative gene panel for model training and evaluation, we implement a two-stage selection process that first filters datasets and then defines the target gene panel. We use data from two public datasets, STImage-1K4M (Chen et al., 2024a) and HEST-1k (Jaume et al., 2024), with spatial gene expressions derived from two sequencing technologies: ST and Visium. The ST data has a capture area of 6.2×6.6 mm, with spots of size 100 µm, and the comparatively higher resolution Visium data has a capture area of 6.5×6.5 or 11×11 mm with spots of size 55 µm.

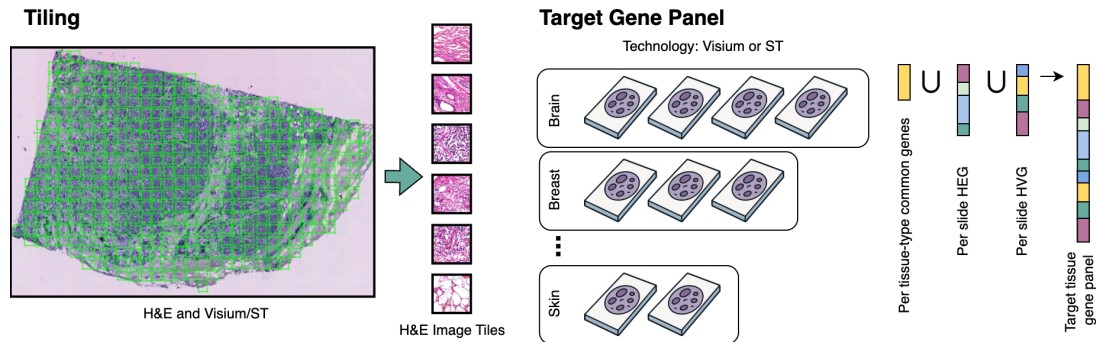

**Figure 1.** Illustration of the gene panel selection strategy. For each tissue-technology cohort, we take the union of genes consistently present across all slides (common genes) and genes that are highly variable or highly expressed (HVGs and HEGs). This results in a cohort-specific gene panel that balances biological diversity with technical robustness.

For each dataset we initially separate the slides based on the technology used. We focus on Visium and ST slides with relatively larger field of view, lower spatial resolution, and higher gene resolution that are more sensible for histomorphology-based gene expression prediction. Within each technology, we further separate the slides according to organ type. Only organ-specific cohorts with more than 40 slides were retained to ensure statistical robustness. For STImage-1K4M, we apply an additional quality control step on the resolution of the H&E-stained images. Since the image resolution varies across slides in this dataset, we retain only Visium slides with a pixel radius of at least 32 pixels, and ST slides with a pixel radius of at least 64 pixels, for sufficient image quality.

After applying these filtering steps, we select six tissue-technology cohorts for our study and define a consistent target gene panel for each, designed to capture both biologically informative and technically robust genes. These panels are constructed by taking the union of the two following gene subsets, as seen in Figure 1.

**Common genes**: The set of genes that are consistently measured across all slides within the cohort.

**Highly variable and highly expressed genes**: For each slide, the top 100 highly variable genes (HVGs) and top 100 highly expressed genes (HEGs) are computed. We then take the union of these 200 genes across all slides within the cohort.

Including common genes and unions over slide-level HVGs and HEGs ensures that the tissue gene panel is not biased toward sample-specific panels, promoting consistency in predictions across diverse tissue types and conditions.

| Dataset | # Slides | # Spots | Gene Panel Size |
|---|---|---|---|
| STImage-ST-Breast | 102 | 45,304 | 8,645 |
| STImage-Visium-Brain | 40 | 775,103 | 4,267 |
| HEST-ST-Brain | 87 | 37,215 | 4,721 |
| HEST-ST-Breast | 108 | 45,305 | 11,440 |
| HEST-ST-Spinal | 302 | 74,104 | 13,923 |
| HEST-Visium-Skin | 71 | 98,188 | 7,643 |

**Table 1.** Overview of selected tissue-technology cohorts and their respective gene panel sizes and spot counts.

### 3.2 Bimodal Dictionary-based Autoencoder

We propose a bimodal autoencoder architecture that leverages shared dictionary-based latent representations for both spatial gene expression profiles and histology image tiles. Each modality is encoded separately into a set of coefficients, which are used to compute a weighted combination of atoms from learnable codebooks—referred to as the *gene codebook* and *image codebook*, respectively. These codebooks serve as modality-specific dictionaries of latent basis vectors that support both inter-modal and intra-modal reconstruction.

By designing the inference method such that image-derived coefficients can be applied directly to the *gene codebook*, we enable a cross-modal reconstruction for predicting gene expression from histology alone. This cross-modality alignment is encouraged during training through a combination of reconstruction losses and latent alignment objectives, resulting in a unified latent space structure that supports interpretable and efficient inference across modalities. The training and inference processes are illustrated in Figure 2.

#### 3.2.1 Gene Branch

The gene branch encodes each log-normalised gene expression vector $x_g \in \mathbb{R}^G$ into a set of latent coefficients $c_g \in \mathbb{R}^K$ via a multi-layer perceptron (MLP) encoder. These coefficients are used to construct a low-dimensional latent representation $z_g \in \mathbb{R}^{d_g}$ through a weighted

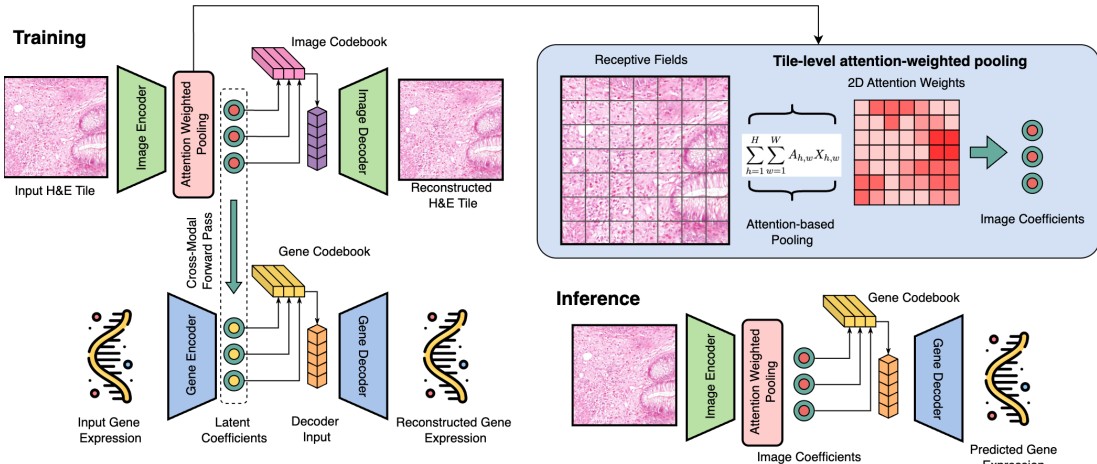

**Figure 2.** Schematic of the bimodal dictionary-based autoencoder architecture and training strategy. The model consists of separate autoencoders for gene expression profiles and histology image tiles, each using a learnable codebook (dictionary) and a set of modality-specific encoders and decoders. During training, both intra-modal (self-reconstruction) and inter-modal (cross-reconstruction) objectives are used, including swapping of latent coefficients from image encoder to gene decoder. The image branch features an attention-based pooling to generate global coefficients. At inference, only the histology image is required to predict gene expression via image encoder, gene codebook, and gene decoder.

combination of $K$ learnable dictionary atoms from a gene codebook $B_g \in \mathbb{R}^{d_g \times K}$. Formally,

$$z_g = B_g c_g = \sum_{k=1}^{K} c_{g,k}\, b_g^{(k)}.$$

The latent vector $z_g$ is then decoded back to gene expression space using an MLP decoder to obtain the reconstruction $\hat{x}_g$. The training objective for this branch is the mean squared error (MSE) reconstruction loss, $\mathcal{L}_g = \|\hat{x}_g - x_g\|_2^2$.

### 3.2.2 Image Branch

The image branch processes histology tiles $x_i \in \mathbb{R}^{3 \times H \times W}$ using a convolutional encoder based on a ResNet-50 backbone, pre-trained on ImageNet (Deng et al., 2009). We choose to use this backbone instead of a foundation model for a fairer, more independent comparison across benchmarks, as discussed in Section 5. The encoder outputs a feature tensor $F \in \mathbb{R}^{C \times 7 \times 7}$. This feature map is split into two sub-branches, as follows.

**Sub-branch 1 (Global Latents)** We apply attention-based pooling over spatial dimensions to produce a global coefficient vector, $c_i \in \mathbb{R}^K$, representing the image in terms of dictionary weights. Specifically,

$$c_i = \sum_{h=1}^{H} \sum_{w=1}^{W} A_{h,w} X_{h,w}$$

where $X_{h,w} \in \mathbb{R}^K$ is the feature vector at spatial location $(h, w)$, and $A_{h,w}$ is the corresponding attention score.

**Sub-branch 2 (Spatial Latents)** A $1 \times 1$ convolutional head outputs spatial coefficient maps $C_i \in \mathbb{R}^{K \times 7 \times 7}$. These coefficients are used to compute local latent features, $Z_i$ via a learnable image codebook $B_i \in \mathbb{R}^{d_i \times K}$:

$$Z_i(:, u, v) = \sum_{k=1}^{K} C_i(k, u, v) \, b_i^{(k)} \in \mathbb{R}^{d_i}, \quad \forall u, v.$$

The resulting spatial latent tensor $Z_i \in \mathbb{R}^{d_i \times 7 \times 7}$ is passed through a symmetric convolutional (transposed convolution) decoder to reconstruct the image $\hat{x}_i$. The reconstruction loss for the image branch is $\mathcal{L}_i = \|\hat{x}_i - x_i\|_2^2$.

### 3.2.3 Cross-modal Autoencoding and Inference

To enable prediction of gene expression directly from histology images, we perform cross-modal decoding by aligning the image latent coefficients with the gene codebook. Specifically, the attention-derived image coefficients $c_i \in \mathbb{R}^K$ are used to select and combine atoms from the gene codebook $B_g \in \mathbb{R}^{d_g \times K}$, producing a cross-modal latent representation:

$$z_{i \to g} = B_g c_i \in \mathbb{R}^{d_g}.$$

This latent vector is then decoded via the gene decoder $D_g$ to yield a predicted gene expression vector $\tilde{x}_g = D_g(z_{i \to g})$. To align the two modalities, we introduce the following loss terms.

**Cross-modal reconstruction loss**

$$\mathcal{L}_{\mathrm{cm}} = \|\tilde{x}_g - x_g\|_2^2,$$

which ensures the predicted gene expression from the image-derived latent vector matches the ground truth.

**Latent alignment loss**

$$\mathcal{L}_{\mathrm{align}} = \|z_{i \to g} - z_g\|_2^2,$$

which encourages the cross-modal latent $z_{i \to g}$ to be close to the encoded gene latent $z_g$.

**Coefficient similarity loss**

$$\mathcal{L}_{\mathrm{coef}} = 1 - \cos(c_i, c_g) = 1 - \frac{c_i^\top c_g}{\|c_i\|_2 \|c_g\|_2},$$

which enforces cosine similarity between the latent coefficient vectors across modalities in the shared coefficient space.

The total training objective is defined as the weighted sum of all reconstruction and alignment components, $\mathcal{L} = \mathcal{L}_i + \mathcal{L}_g + \lambda_{\mathrm{cm}} \mathcal{L}_{\mathrm{cm}} + \lambda_{\mathrm{align}} \mathcal{L}_{\mathrm{align}} + \lambda_{\mathrm{coef}} \mathcal{L}_{\mathrm{coef}}$, where we set $\lambda_{\mathrm{cm}} = 1.0$, $\lambda_{\mathrm{align}} = 0.1$, and $\lambda_{\mathrm{coef}} = 0.1$ in all experiments, ensuring the end to end training for each modality whilst allowing for learned alignment in the latent space.

**Inference** At inference time, only the histology image $x_i$ is required. The image encoder produces the coefficient vector $c_i$, which is then used to generate the gene latent vector via the gene codebook, $\tilde{x}_g = D_g(B_g c_i)$, yielding a single forward-pass prediction of the spatial gene expression at a given location. This design enables efficient, scalable inference without requiring reference retrieval or post-processing.

## 4 Experiments

### 4.1 Preprocessing

Preprocessing of the gene expression data is a critical step to ensure the comparability and interpretability of the input-output mappings during model training. For the spatial transcriptomics data from each selected tissue-technology cohort, we follow a consistent and minimal preprocessing pipeline inspired by standard single-cell transcriptomics workflows, resulting in cohort-specific gene panels with sizes ranging from approximately 4,000 to over 13,000 genes as seen in Table 1. For the images, we use a 224×224 crop around each spot, and apply standard augmentation methods. Full details can be found in Appendix A.

At the slide-level we split each dataset into training (70%), validation (10%), and test (20%) sets, repeating this process randomly five times for different folds.

### 4.2 Results

**Metrics** To comprehensively evaluate predicted gene expression from H&E, we report the following metrics: *L1 Error*, mean absolute difference between predicted and true expression values, averaged over all genes and spots; *Pearson Correlation*, correlation across the entire gene panel, reflecting overall linear association; *Spearman Rank Correlation*, correlation within the top 50, 200, and 1000 most highly expressed genes, capturing the preservation of gene expression ordering; *Recall at Top-k*, fraction of truly top-$k$ expressed genes recovered in the model's top-$k$ predictions for each spot.

**Benchmarks** We compare our proposed method to other qualified methods in the literature, such as established methods ST-Net (He et al., 2020) and BLEEP (Xie et al., 2023), which have shown good results in more than one publication or competition. We implement DeepSpot (Nonchev et al., 2025), incorporating all spot, subspot and neighbourhood regions, the latter defined by a radius of 3 spots. We use the UNI foundation model (Chen et al., 2024b) to encode the images in the DeepSpot implementation, for better comparison with our benchmark of an MLP trained on top of the frozen UNI features (UNI-MLP). We implement CLIP (Radford et al., 2021) with our GenST autoencoder models as an alternative method to align the modalities in the latent space, as used also in the mclSTExp model (Min et al., 2024). We also compare the results to the gene-only VQ-VAE from our GenST model, titled gene2gene. All models were trained for 100 epochs using the RAdam optimizer (Liu et al., 2020) with learning rate 0.0001 and weight decay 0.0001.

Results for our primary metric, mean Pearson correlation across the five folds, are given in Table 2. Full results for all metrics are visualised in Appendix B. Our GenST model outperforms other benchmarks in 3/6 datasets, with the UNI-MLP model performing best in the other 3/6 datasets. No models reach the performance of the gene2gene autoencoder, showing that there is more information in the spatial transcriptomics data that we cur-

rently cannot extract from the histology image with the methods studied here. We also reason that our Pearson correlation scores, superior across benchmark models compared to other experiments in the literature, are due to our Target Gene Panel Selection method in Section 3.1.

| Dataset | ST-Net | BLEEP | CLIP | DeepSpot | UNI-MLP | GenST | gene2gene |
|---|---|---|---|---|---|---|---|
| **STImage** | | | | | | | |
| ST-Breast | 0.757±0.01 | 0.640±0.01 | 0.715±0.01 | 0.752±0.01 | 0.743±0.01 | **0.763**±0.01 | 0.832±0.01 |
| Visium-Brain | 0.785±0.07 | 0.542±0.05 | 0.609±0.35 | 0.800±0.09 | 0.775±0.11 | **0.804**±0.08 | 0.922±0.02 |
| **HEST** | | | | | | | |
| ST-Brain | 0.908±0.01 | 0.894±0.01 | 0.904±0.00 | 0.913±0.00 | **0.913**±0.00 | 0.905±0.00 | 0.933±0.00 |
| ST-Breast | 0.573±0.08 | 0.529±0.04 | 0.582±0.03 | 0.635±0.04 | **0.642**±0.06 | 0.568±0.07 | 0.854±0.01 |
| ST-Spinal | 0.731±0.01 | 0.689±0.01 | 0.720±0.01 | 0.726±0.01 | **0.737**±0.01 | 0.732±0.01 | 0.776±0.01 |
| Visium-Skin | 0.699±0.09 | 0.522±0.07 | 0.585±0.25 | 0.603±0.13 | 0.720±0.07 | **0.724**±0.06 | 0.835±0.03 |

**Table 2.** Mean Pearson correlation with standard deviation over five test folds from all models on all dataset cohorts.

## 5 Discussion

We develop a robust and interpretable approach to predicting spatial transcriptomics expressions from the histology image, effectively conserving and combining information from both modalities. We also perform clustering analysis on our predictions, as seen in Appendix C, which demonstrates the spatial robustness of our GenST model.

While the UNI-MLP also performed well in our experiments, it should be noted that the UNI pathology foundation model (Chen et al., 2024b) was released by the same research group as the HEST-1k dataset (Jaume et al., 2024). The foundation model was trained partly on unspecified internal hospital data while the HEST-1k dataset was also collected from internal data cohorts, suggesting potential similarities between images in the HEST-1k dataset and those seen in training the UNI model. This would skew comparative performance of the UNI model on the HEST-1k dataset here, which is used in both the UNI-MLP and DeepSpot benchmarks.

To develop this work further, we suggest experimenting with foundation models for both the image and gene encoders (Wang et al., 2025a), curating a list of genes relevant to the clinical setting, trying different approaches for the cross modal computation in the latent space and domain adaptation across data cohorts. To better enable this, we make our code publicly available at `https://github.com/ox-ibme-bio-imaging/GenST-workshop-version`.

## Acknowledgments and Disclosure of Funding

RW acknowledges funding from the EPSRC Doctoral Prize. Views expressed are those of the authors and not necessarily those of EPSRC, UKRI and Innovate UK.

## Appendix A. Data Processing

First, for the spatial transcriptomics data, we applied total count normalization across each spot, followed by log transformation to stabilize variance and reduce the dynamic range of the data. This preprocessing ensures that gene expression values are on a comparable scale across spots and slides, enabling effective training of both gene autoencoders and cross-modal prediction models. Following normalization, we constructed the target gene panel for each cohort as described in Section 3.1. This resulted in a cohort-specific set of genes used as prediction targets, allowing the model to flexibly adapt to differences in gene coverage across datasets while maintaining meaningful biological representation.

While many existing approaches in this field restrict training and evaluation to the top 250 or 1000 highly variable or highly expressed genes (Nonchev et al., 2025; Zeng et al., 2022; Min et al., 2024; He et al., 2020), thereby excluding sparsely measured genes, we adopt a more comprehensive strategy. For each tissue-technology cohort, we construct a cohort-specific gene panel that includes both biologically informative (highly variable and highly expressed) genes and genes commonly measured across slides. This results in panel sizes ranging from approximately 4,000 to over 13,000 genes, enabling a more challenging yet thorough evaluation setting. Our approach facilitates the potential discovery of novel biomarkers, including those with lower expression levels that are often omitted in prior work.

For the corresponding histology image, we use an image crop around each spot location and resize to 224×224 if required. The following augmentations (from the torchvision.transforms.v2 Python library) were used on these images in training: a random vertical flip with probability 0.5; a random horizontal flip with probability 0.5; colour jitter with brightness 0.02, contrast 0.05, saturation 0.1 and hue 0.1; random sharpness adjustment with sharpness factor 0.2 and probability 0.2; random autocontrast with probability 0.5; random rotation by multiples of 90 degrees. Additionally, as on the validation and test datasets, we normalised the images per channel by mean (0.485, 0.456, 0.406) and standard deviation (0.229, 0.224, 0.225).

## Appendix B. Results

Full results for all metrics are visualised in Figure 3. Each metric captures a different aspect of model performance. L1 error is sensitive to prediction scale and magnitude. Pearson correlation measures global linear trends but is dominated by highly expressed genes. Spearman and Top-$k$ recall are rank-based, better reflecting relative ordering and recovery of key genes, but do not account for absolute error. Reporting all these metrics ensures fair and complete assessment of both quantitative accuracy and biological relevance.

Preprocessing of the gene expression counts is a crucial stage for successful predictions with most models, and that the scaling of these counts is an important step that can be done differently. For example, taking the log transform of the counts is standard, but these can then be scaled by a factor, ranging anywhere from 10,000-1,000,000 (Nonchev et al., 2025; Min et al., 2024). Therefore, we provide the L1 metric for comparison between models presented here, for which the data was processed identically.

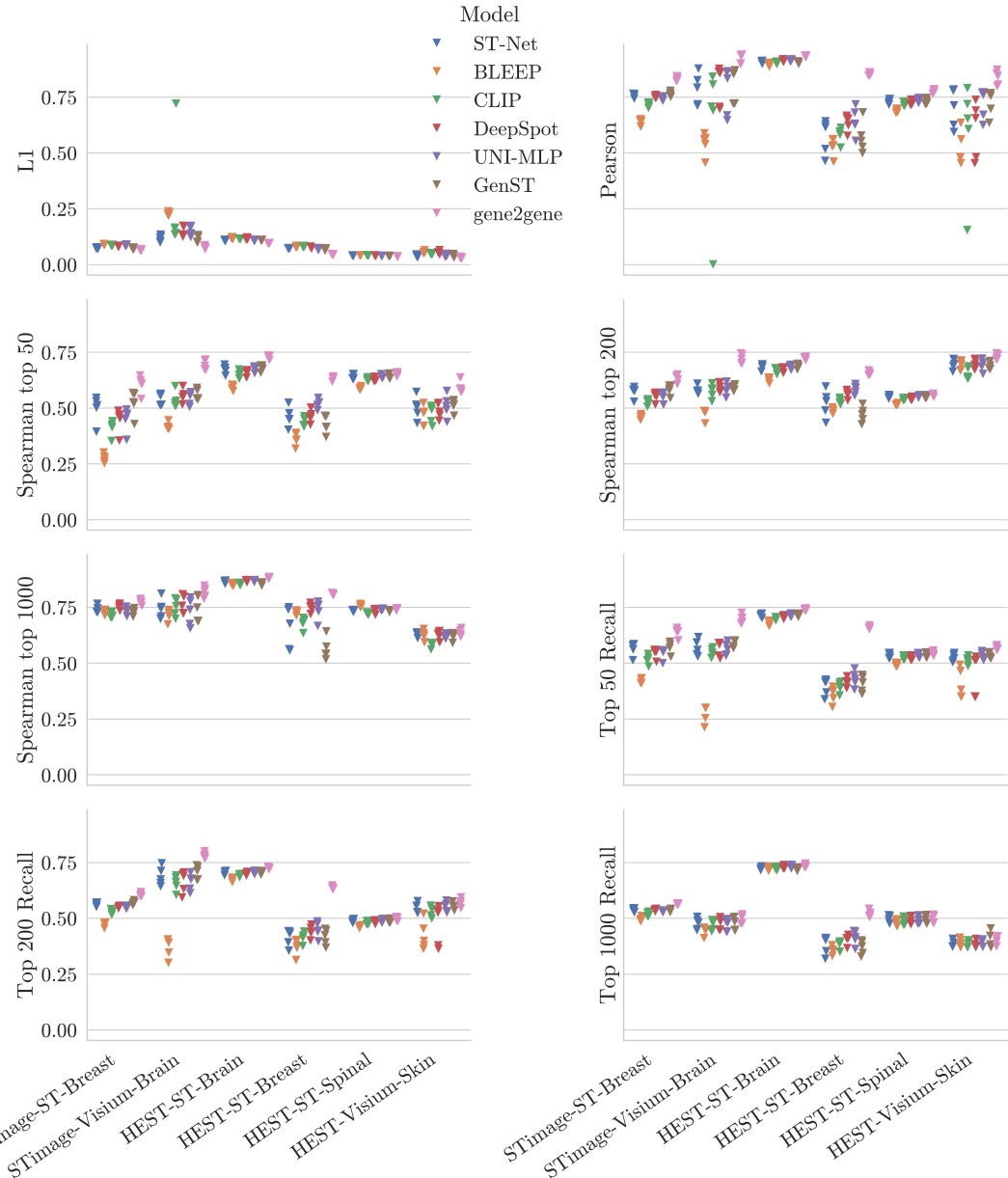

**Figure 3.** Test results shown per metric, coloured by model and grouped into columns by tissue-technology cohort, shown over all five folds to demonstrate the variance in results. Note the gene2gene model is the gene-only autoencoder part of our model, GenST, and is provided as a proxy for the best metrics that can be achieved from our cross-modal approach.

## Appendix C. Clustering

| Dataset | Metric | DeepSpot | GenST | ST-Net |
|---------|--------|----------|-------|--------|
| STimage-ST-Breast | Hungarian Accuracy | 0.477 (0.010) | **0.497** (0.015) | 0.486 (0.011) |
| | ARI | 0.172 (0.014) | **0.190** (0.023) | 0.179 (0.020) |
| | NMI | **0.240** (0.016) | 0.231 (0.015) | 0.201 (0.014) |
| STimage-Visium-Brain | Hungarian Accuracy | **0.363** (0.004) | 0.361 (0.008) | 0.343 (0.006) |
| | ARI | 0.164 (0.010) | **0.179** (0.006) | 0.166 (0.005) |
| | NMI | 0.235 (0.014) | **0.248** (0.011) | 0.235 (0.008) |
| HEST-ST-Brain | Hungarian Accuracy | **0.529** (0.007) | 0.518 (0.007) | 0.515 (0.007) |
| | ARI | 0.271 (0.011) | **0.279** (0.009) | 0.277 (0.012) |
| | NMI | **0.368** (0.008) | 0.349 (0.007) | 0.358 (0.009) |
| HEST-ST-Breast | Hungarian Accuracy | 0.485 (0.011) | 0.488 (0.013) | **0.488** (0.009) |
| | ARI | **0.196** (0.017) | 0.191 (0.022) | 0.182 (0.011) |
| | NMI | **0.253** (0.012) | 0.249 (0.015) | 0.224 (0.010) |
| HEST-Visium-Skin | Hungarian Accuracy | **0.414** (0.003) | 0.393 (0.005) | 0.388 (0.005) |
| | ARI | **0.159** (0.005) | 0.149 (0.006) | 0.140 (0.005) |
| | NMI | **0.261** (0.010) | 0.253 (0.008) | 0.221 (0.006) |

**Table 3.** Mean clustering metric scores with variance provided in brackets across datasets.

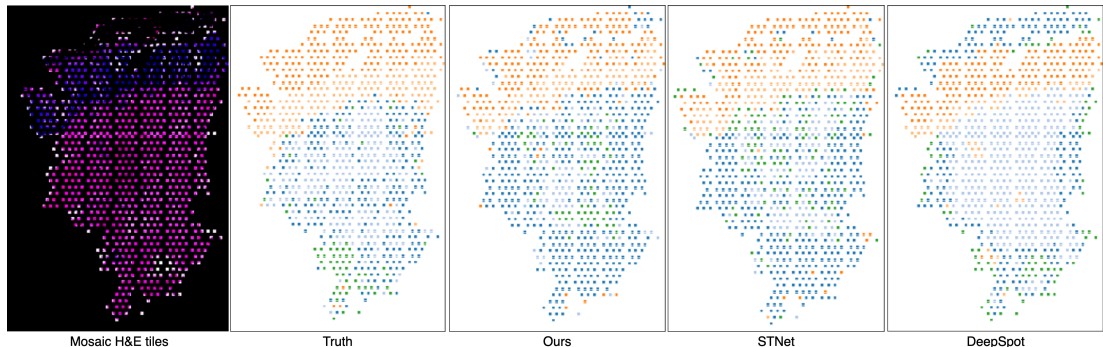

**Figure 4.** Example spatial clustering results on HEST-Visium-Skin. From left to right: (1) H&E mosaic image reconstructed from tiles, followed by clusters from (2) Ground truth gene expression, (3) GenST predictions, (4) ST-Net predictions, (5) DeepSpot predictions. Cluster assignments are visualized as distinct colors and mapped back to their original spatial locations, illustrating that GenST recovers spatial domains closely matching the ground truth structure.

To assess whether our model's predicted gene expression profiles preserve meaningful spatial and biological structure, we performed unsupervised clustering on the predicted expression matrices and compared the resulting clusters to those derived from ground-truth gene expression. For each tissue-technology cohort, clustering was performed using Leiden algorithm on both true and predicted gene expression, and the results were mapped back to the original spatial locations. Cluster agreement was evaluated using Hungarian Accuracy

(i.e. the optimal matching accuracy after aligning cluster assignments between methods using the Hungarian algorithm), Adjusted Rand Index (ARI), and Normalized Mutual Information (NMI), as reported in Table 3.

Across all cohorts, GenST achieves competitive clustering metrics, with the highest ARI in 3 out of 5 settings (Table 3), indicating that the predicted gene profiles largely capture the underlying spatial organization. DeepSpot, which leverages neighborhood information, often yields stronger spatial clustering structure, although this does not necessarily translate to higher spot-level prediction accuracy (as shown in previous evaluation results). Visualizations in Figure 4 show that clusters inferred from predictions are broadly aligned with those from ground truth expression.

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
