# OpenReview forum: "GenST: A Generative Cross-Modal Model for Predicting Spatial Transcriptomics from Histology Images"
_MICCAI.org/2025/Workshop/COMPAYL — COMPAYL 2025_

### Official Review · Reviewer_BU3G · 2025-07-04
**A promising generative cross-modal framework for spatial transcriptomics prediction from histology images**

**Rating:** 4
**Confidence:** 3

**Review:**

1. Short Summary

This paper introduces GenST, a novel generative framework that predicts spatial transcriptomics from H&E-stained histology images. By aligning the latent spaces of two Vector Quantized Variational Autoencoders (VQ-VAEs), one for gene expression and one for image patches: the model achieves interpretable and robust cross-modal generation. Extensive experiments across two large-scale public datasets (STImage-1K4M and HEST-1k) demonstrate that GenST achieves competitive predictive performance against state-of-the-art models, including ST-Net, DeepSpot, and BLEEP.

2. Strengths

Well-motivated problem: The cost and complexity of ST technologies make computational prediction a highly valuable research direction;
Interpretable architecture: The use of VQ-VAE with dictionary-based latent space enables semantic control and interpretability across modalities;
Rigorous evaluation: The authors benchmark the model across six tissue-technology cohorts with meaningful gene panel selection and robust preprocessing；Diverse and relevant metrics: Evaluation includes Pearson/Spearman correlation, L1 error, Recall@Top-k, and clustering metrics (ARI, NMI), providing a comprehensive view of performance；Scalable inference: The model supports one-shot prediction from histology without requiring reference retrieval, which is practical for real-world deployment.

3. Weaknesses

Incremental novelty: The approach builds upon existing methods like BLEEP and CLIP-aligned models; while technically well-executed, the conceptual advancement is moderate；Limited ablation analysis: The paper does not deeply examine the contribution of different loss terms or encoder architectures；Information gap not well explained: The image-based models significantly underperform the gene-only autoencoder (gene2gene), which deserves more in-depth discussion.

4. Detailed Comments

The authors acknowledge potential data overlap between the UNI model and the HEST-1k dataset, which may inflate UNI-MLP benchmark results. This should be better controlled or discussed.
A UMAP or t-SNE visualization of the shared latent space could help illustrate the cross-modal alignment capability of GenST.
The design choice of VQ-VAE is well-argued, but a small comparison to continuous latent methods (e.g., standard VAE) would further contextualize performance.
Model training complexity and resource consumption (e.g., GPU hours, memory footprint) are not reported and would aid practical adoption assessment.
Since the model can generalize across datasets and tissues, it could be interesting to explore transfer learning or domain adaptation in future work.

---

### Official Review · Reviewer_KioS · 2025-07-13
**Generative cross-modal alignment improves organ-specific gene expression prediction from histology**

**Rating:** 5
**Confidence:** 4

**Review:**

**Summary**: The manuscript proposes a method to predict gene expression from H&E tissue patches trained on spatial transcriptomics data. ST samples from STImage-1K4M and HEST-1K are split into technology and organ-specific cohorts with each at least 40 slides resulting in six cohorts. For every technology-organ cohort, a different gene panel is composed out of the union of the top-100 highly varying and top-100 highly expressed genes per slide and the common genes that are measured for all slides in the cohort. The method is based on two VQ-VAEs, using ResNet50 as image backbone and an MLP as gene backbone. Each autoencoder is trained for self-reconstruction and, additionally, the latent spaces are aligned for reconstruction of gene expression from images at inference.

**Strengths**:
- Well-written and structured
- Represents the state-of-the-art accurately
- Well-motivated method including gene selection and alignment and reconstruction objectives

**Weaknesses**:
- Although I acknowledge that this is mentioned in the manuscript's outlook, it remains unclear to me why current pathology foundation models are not used as image backbones. Especially, as the comparison method UNI-MLP shows very good performance.
- The effect of the local and global branches and the choice of VQ-VAEs are not quantified.
- There is no standard deviation given to assess the statistical effects.
- Code and details on the samples contained in the six training cohorts are missing.

**Minor issues**:
- Please, add more concrete information on results of the paper to the abstract, such as dataset names and sizes, organs involved, downstream evaluations, and findings.
- Table 2: Think about sorting the methods by date of appearance instead of alphabetically. This makes it easier to compare the models as this also reflects the semantic order mentioned in the main text.
- Replace “x” with “\times”
- It would be nice to slightly increase the font size in all figures as there is enough white space and it is tedious to read in the current format.
- p. 7: can’t > cannot
- Figure 4: “Ground truth”

---

### Official Review · Reviewer_X5hs · 2025-07-15
**Interesting paper, more experiments needed**

**Rating:** 4
**Confidence:** 5

**Review:**

This work proposes a generative model (GenST) that predicts spatial transcriptomics from H&E-stained histology images by aligning the latent spaces of two VQ-VAEs. The method demonstrates improved performance on some datasets and offers an interpretable framework linking tissue morphology to gene expression. Overall it is an interesting approach for predicting spatial transcriptomics from histology images. The work would be significantly strengthened by more thorough ablation studies and empirical validation of key components.


Pros:

The manuscript is clear and generally well written.

The proposed method is intuitive, well-structured, and implemented in an elegant manner.

Major Concerns:

The paper lacks sufficient ablation studies to support the proposed design choices.

For instance, the model is trained with multiple loss terms, each weighted differently. However, the impact of each component and the sensitivity to the choice of λ values are not explored.

The proposed Target Gene Panel Selection, which is highlighted, is not clearly demonstrated to be superior to existing methods.

Minor Concerns:

Although the method shows improvements on some datasets, the baseline UNI-MLP outperforms GenST on half of them.

Other Comments:

It would be nice if the manuscript could be re-read to fix some language errors.

Code, Model Weights, and/or Data Availability:

The GitHub repository was not available at the time of review.